# Preparation of Fe^3+^ Doped High-ordered TiO_2_ Nanotubes Arrays with Visible Photocatalytic Activities

**DOI:** 10.3390/nano10112107

**Published:** 2020-10-23

**Authors:** Jin Zhang, Chen Yang, Shijie Li, Yingxue Xi, Changlong Cai, Weiguo Liu, Dmitriy Golosov, Sergry Zavadski, Siarhei Melnikov

**Affiliations:** 1Shaanxi Province Key Laboratory of Thin Films Technology and Optical Test, School of Optoelectronic Engineering, Xi’an Technological University, Xi’an 710032, China; yangchen931@163.com (C.Y.); lishijie@xatu.edu.cn (S.L.); xiyingxue@163.com (Y.X.); changlongcai@126.com (C.C.); wgliu@163.com (W.L.); 2Belarusian State University of Informatics and Radioelectronics, Electronic Technique and Technology Department, Center 10.1, Thin Film Research Laboratory, 6 P. Brovka str., 220013 Minsk, Belarus; dmgolosov@gmail.com (D.G.); szavad@mail.ru (S.Z.); snmelnikov@gmail.com (S.M.)

**Keywords:** Fe^3+^ doping, MO degradation, TiO_2_ nanotubes, visible photocatalysis

## Abstract

In this paper, the Fe^3+^ doped rutile phase TiO_2_ nanotubes arrays (NTAs) were prepared in a low temperature water-assistant crystallization method. It is noteworthy that the Fe^3+^ doping hardly hinders either the crystallization of rutile TiO_2_ NTAs or the highly-ordered nanotubular morphologies. Moreover, Fe^3+^ did not form other compound impurities, which indicated that Fe^3+^ substitute Ti^4+^ into the lattice of TiO_2_. With the introduction of Fe^3+^, the light absorption range of TiO_2_ NTAs extends from the ultraviolet band to the visible light range. Photocatalytic testing results indicate that Fe^3+^ doped TiO_2_ NTAs can effectively improve the degradation rate of methyl orange aqueous solution in visible light, and the TiO_2_ NTAs with 0.2 mol/L Fe^3+^ doping exhibits the highest photocatalytic degradation efficiency.

## 1. Introduction

TiO_2_, as one of the most important derivatives of titanium, has been widely used in photocatalytic degradation of organic pollutants, hydrogen generation by water splitting, solar cells, lithium-ion batteries, sensors, and so on. This is due to its excellent photochemical activities, good electronic properties, low cost, and being non-toxic to environments [1,2,3,4,5,6,7]. Compared with TiO_2_ nanoparticles, TiO_2_ nanotube arrays (NTAs), as a fast electron transport channel, have attracted much attention in the fields of photocatalysis and solar cells [8,9,10]. However, as a photocatalyst, TiO_2_ nanotubes and nanoparticles share the same defects. Their wide band gap makes them insensitive to visible light, which greatly reduces the utilization efficiency of sunlight [11,12,13,14]. To solve this problem, various approaches have been proposed to improve the photo-response range of TiO_2_ materials [15,16,17,18,19], among which metal ion doping is one of the most common and promising methods [20,21,22,23,24,25,26,27,28].

Metal ion doping modification mainly includes 3d transition metal ions into the crystals of titanium dioxide, taking advantage of the instability of the outer electrons of 3d transition metals [29,30,31]. On one hand, when these unstable and unsaturated electrons are presented inside titanium dioxide, it helps to reduce the recombination rates of photogenerated electron-hole pairs on the surface of TiO_2_ and improve the utilization efficiency of photogenerated electrons [32,33,34,35]. On the other hand, some metal ions would form TiO_2_ electron traps in the band gap, and then expand the photo response of TiO_2_ NTAs to visible light [36,37]. Among various transition metal ions, Fe^3+^ has demonstrated excellent properties when doped with TiO_2_ NTAs [26,38,39]. However, preparation and usage of Fe^3+^ doped TiO_2_ NTAs reported in the literature does not achieve low power consumption, low cost and effective recycling. For example: Sun et al. used a high temperature annealing process to prepare TiO_2_ NTAs [26], Yu et al. adopted a high temperature alloy process to fabricate the Fe doped Ti foil [38], and Pang et al. prepared discrete Fe^3+^ doped TiO_2_ NTAs, which is not conducive to rapid recycling [39].

In order to facilitate recycling, we proposed a simple and low-cost preparation process of Fe^3+^-doped TiO_2_ NTAs on Ti foil under mild environment. In this study, anodic TiO_2_ NTAs were crystallized at low temperature induced by Fe^3+^ aqueous solution and the conditions and preparation process were optimized through multiple experiments. Various characterization and experimental results indicate that the prepared Fe^3+^ doped TiO_2_ NTAs have good photocatalytic activities for the degradation of organic substance under visible light irradiation.

## 2. Materials and Methods

The initial amorphous TiO_2_ NTs were synthesized by electrochemical anodization of Ti foil (1.6 cm × 3 cm, >99.8%, 0.25 mm, Alfa Aesar, Shanghai, China) in glycol containing electrolyte (490 mL glycol (Alfa Aesar), 10 mL H_2_O and 1.66 g NH_4_F (Alfa Aesar)) at room temperature under a constant voltage of 50 V in a two-electrode configuration (a Ti foil as the working electrode and a platinum foil as the counter electrode), using a DC programmable power supply (Model IT6154, ITECH Electronics Co., Ltd. Nanjing, China). After anodization for 1.5 h, highly ordered TiO_2_ nanotube arrays on the Ti sheet substrate were obtained. Then, the as-prepared TiO_2_ NTs were thoroughly washed with a large amount of anhydrous alcohols and dried in air.

The Fe^3+^ doped TiO_2_ NTs were prepared by the water induced low temperature crystallization technology. The amorphous titanium dioxide nanotube films were immersed in mixed solutions which contained an equal volume of water and absolute alcohol, and 0.05 mol/L, 0.10 mol/L, 0.20 mol/L FeCl_3_ (Alfa Aesar) were added into the solvent, separately. Then, the mixed solutions were heated to 70 ^o^C and held for 10 h to fully crystallize the TiO_2_ NTAs. After that, the samples were dried naturally at 20 ^o^C room temperature to ensure the stability of the nanotube. It must be mentioned that we also tried to prepare higher doping concentration samples, but the TiO_2_ NTAs layers broke into pieces due to high internal stress caused by the excessive doping concentration.

X-ray diffraction (XRD) analysis (D8 ADVANCE, Bruker AXS, Karlsruhe, German) was employed to characterize the crystal structure and composition of the samples. The X-ray radiation source was Cu Kα, obtained at 40 kV, 30 mA, the scanning speed was 3.6°/min at a step of 0.03°, and the range of 2θ is 20–80°. The morphological property and energy dispersive spectrometry (EDS) were observed with a field-emission scanning electron microscopy (GeminiSEM 500, Zeiss, Aalen, German). Transmission electron microscopy (TEM) and EDS mapping were performed on a transmission electron microscope (JEOL JEM-2100). Raman testing and analysis were used a laser Raman spectroscope (LabRAM HR Evolution, HORIBA, Fukuoka, Japan) with a 532 nm line for the excitation. The absorption spectra were collected by using integrating sphere accessory of the UV−vis spectrometry (Ultrospec 2100 pro, Harvard Apparatus, Cambridge, MA, USA).

The photocatalytic activities of Fe^3+^ doped TiO_2_ NTAs were evaluated on the basis of the degradation of methyl orange (MO) as model organic pollutants in aqueous solutions and measured by the UV-vis spectrometer (JASCO V-570 UV/vis/NIR, JASCO, Tokyo, Japan). The surface area (3.0 × 1.6 cm^2^) of the samples were immersed in the 5 × 10^−5^ mol/L MO aqueous solution and irradiated with eight 4W UV bulbs (Toshiba, Black Light Blue, FL4W, Tokyo, Japan), eight 4W visible bulbs (Toshiba, Cool white, FL4W, Tokyo, Japan) with UV filter (>480 nm), for UV photoactivities test and visible photoactivities test, respectively. Before measurement, the samples were soaked for 30 min in the dark with magnetic stirring to reach adsorption/desorption equilibrium. The photo-degradation measurement was carried out in an open quartz photoreaction vessel with rapid stirring for 120 min under room temperature. The concentration of the residual MO was measured by the UV-vis spectrometer at about 460 nm. The photocatalytic performance data of each sample were tested for 10 times, and the mean value and error were calculated and plotted.

## 3. Results and Discussion

Figure 1 shows a schematic synthesis procedure of Fe^3+^ doped TiO_2_ NTAs. In step 1, self-organized TiO_2_ NTAs were firstly prepared by electrochemical anodization. A yellowish film could be observed, which is attributed to the amorphous TiO_2_ NTAs coated on the Ti foil. Then the as-prepared amorphous TiO_2_ NTAs film was immersed into aqueous solution of Fe^3+^ at 70 °C to induce crystallization and doping. After this step, the film turned a reddish color, which could be attributed to the doping of Fe^3+^ ion in TiO_2_ NTAs.

In order to detect the effect of Fe^3+^ doping on the morphology of TiO_2_ NTAs, SEM was used to observe and characterize the morphology of titanium dioxide nanotubes. As can be observed from Figure 2a, the TiO_2_ NTs are top-opened and have an inner diameter of about 80–90 nm with a wall thickness of ~20 nm. After doping with Fe^3+^, no significant difference in the nanotube diameter can be observed, while the wall thickness gets slightly increased about 2–5 nm (see Figure 2b). The cross section of Fe^3+^ doped TiO_2_ NTAs is exhibited in Figure 2c; it can be seen that the inner surface of the nanotubes is clean and not blocked by particles. The SEM test results show that the crystallization/doping process did not damage the top-open and highly oriented morphologies of TiO_2_ NTAs, which is advantageous for the organic solvent to be fully filled into the nanotubes during the photocatalytic process. The weight ratios of Ti, O, and Fe in samples with different doping concentrations were also tested by using EDS, and the results were presented in Table 1. It can be observed that with the increase of Fe^3+^ doping concentration, the Fe content gradually rises, the oxygen content basically remains unchanged, and the Ti content gradually decreases in these samples, indicating that the Fe^3+^ is likely substituting Ti^4+^ into the lattice of TiO_2_.

For the sake of observing the distribution of Fe^3+^ in TiO_2_ nanotube more intuitively, TEM and EDS mapping were used to test the distribution of Ti, O, and Fe elements in a single 0.2 mol/L Fe^3+^ doped TiO_2_ nanotube as shown in Figure 3. We can observe that there is no cladding layer and enrichment of the Fe element is not present on the surface of the TiO_2_ nanotube, and the Fe element uniformly distributes in the TiO_2_ nanotube, indicating that the Fe did not form compounds coated on the surface of TiO_2_ nanotube.

The XRD patterns of TiO_2_ NTAs samples with different Fe^3+^ doping concentrations are shown in Figure 4. It can be seen that the diffraction peaks of the three samples are quite similar. After comparison with standard PDF cards, we find that these main diffraction peaks well match (110), (101), (211), (002), (311) planes of rutile phase TiO_2_ and (100), (002), (101) planes of titanium. In addition, no other impurity peaks are found, and whether Fe^3+^ is doped into TiO_2_ nanotubes in alternative forms needs further discussion. The lattice parameters and interplanar crystal spacing (d values) of pure and Fe^3+^ doped TiO_2_ NTAs samples are outlined in Table 2. With the increase of Fe^3+^ doping concentration, the lattice constant and d values decrease slightly; the main reason is that the radius of Fe^3+^ (64 Å) is slightly smaller than that of Ti^4+^ (68 Å). This can also be proved from the side that Fe^3+^ replaces Ti^4+^ and enters into the lattice of TiO_2_.

To further verify that Fe^3+^ did not form a compound but entered into the TiO_2_ lattice, the Raman spectrum of the 0.2 mol/L Fe^3+^ doped sample was further investigated and analyzed. The Raman spectrum and peak fitting curves of the sample before photocatalytic experiments are shown in Figure 5a. After fitting the spectral line, we find that there are only three Raman peaks of rutile phase TiO_2_ (234 cm^−1^ (Eg mode), 445 cm^−1^ (Eg mode), and 609 cm^−1^ (A1g mode)) [26] and no other impurity peaks exist, which also confirmed the XRD test results that Fe was doped into the lattice of TiO_2_ and no impurity was formed.

Figure 6 shows the UV-Vis absorption spectra of the pure and Fe^3+^ doped TiO_2_ NTAs. As can be seen from the spectra, the TiO_2_ NTAs samples without Fe^3+^ doping shows an absorption characteristic only in the ultraviolet region (<380 nm), but not in the visible region. Once doped with Fe^3+^, the red shift of absorption edge can be observed clearly in the Figure 6. The absorption capacity of the ultraviolet region was also significantly enhanced. This result, along with the XRD and Raman analysis above, reveal that the Fe^3+^ ions were indeed introduced into the lattice of TiO_2_ and formed the impurity energy level, thus expanding the absorption spectrum range of TiO_2_ nanotubes.

The photocatalytic activities of samples were measured by degradation of methyl orange (MO). Figure 7 shows the concentration change of MO as a function of irradiation time under ultraviolet light for pure and doped TiO_2_ NTAs with a different amount of Fe^3+^. First, we find that the photocatalytic activity of the sample doped with Fe^3+^ in 0.05 mol/L precursor solution can only degrade MO about 33% in 2 h, which is much lower than that of pure titanium dioxide nanotubes. This is might due to the fact that the UV light has provided enough energy to generated electron-hole pairs, and a small amount doping of Fe^3+^ tends to act as a recombination center, which will decrease the photoactivities [40]. However, with the increase of the Fe^3+^ concentration, the catalytic activity increased gradually and reached the best level when the concentration was 0.2 mol/L. Under this condition, the major effect of the Fe^3+^ doping is to introduce impurity levels in the band gap of TiO_2_, and become a shallow trap of photogenerated electrons or holes, for reducing the recombination of electron-hole pairs [41].

For practical application, it is necessary to estimate the photocatalytic activity of these samples under visible light irradiation, which is shown in Figure 8. The results clearly revealed that doping with Fe^3+^ does greatly enhance the photochemical activity of TiO_2_ NTAs under visible light. Photogenerated charge carriers successfully flow to Fe^3+^ energy level and form an electron trap which shortens the distance between conduction band and valence band [42,43]. The successful doping of Fe ion into TiO_2_ makes it more liable for the visible radiation to excite the electrons in conduction band of TiO_2_. Along with the characterization and experiment above, it can be concluded that TiO_2_ NTAs have the best photocatalytic performance doped with 0.2 mol/L of Fe^3+^.

After photocatalytic experiments, the Raman spectrum and EDS of 0.2 mol/L Fe^3+^ doped TiO_2_ NTAs were also investigated. Figure 5b shows the Raman spectrum and peak fitting curves of the sample after photocatalytic experiments. Still, only three characteristic peaks located at 236, 447, and 610 cm^−1^ can be observed, which is consistent with the Raman results before the photocatalytic experiments. In addition, after photocatalytic experiments the weight ratios of Ti, O, and Fe in the sample are 41.01, 50.23, and 8.76% w/w, respectively, which also coincides with the original results. The above test results indicate that Fe^3+^ did not form other compound impurities and could not be released in the solution during the photoirradiation.

## 4. Conclusions

In summary, Fe^3+^ doped rutile phase TiO_2_ NTAs have been successfully prepared by a simple and low-cost method which combined electrochemical anodization and low temperature water-assistant crystallization. The results of test and analysis show that Fe^3+^ doping not only doesn’t change the crystallization of TiO_2_ NTAs, but also no Fe-containing compounds impurities are produced. The Fe^3+^ substitutes Ti^4+^ into the lattice of TiO_2_ and forms the impurity energy level, which expands the absorption range of TiO_2_ NTAs from ultraviolet to the visible band. It is noteworthy that Fe^3+^ doping significantly improves the photocatalytic activity of TiO_2_ NTAs in the visible light region, making it have a promising practical application prospect in the photocatalytic decomposition of organic pollutants.

## Figures and Tables

**Figure 1 nanomaterials-10-02107-f001:**
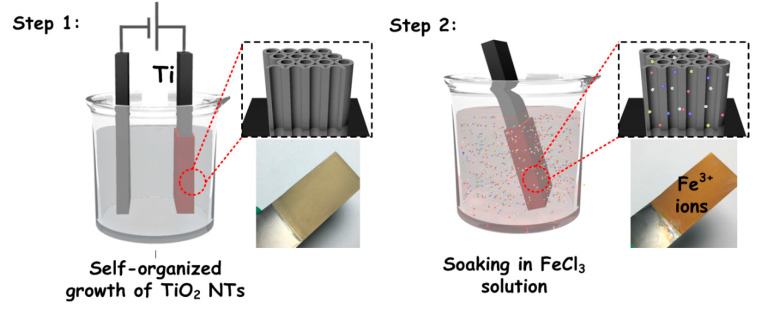
Schematic synthesis diagram of the amorphous TiO_2_ and Fe^3+^-doped TiO_2_ nanotubes arrays (NTAs), involved the photographs of one sample.

**Figure 2 nanomaterials-10-02107-f002:**
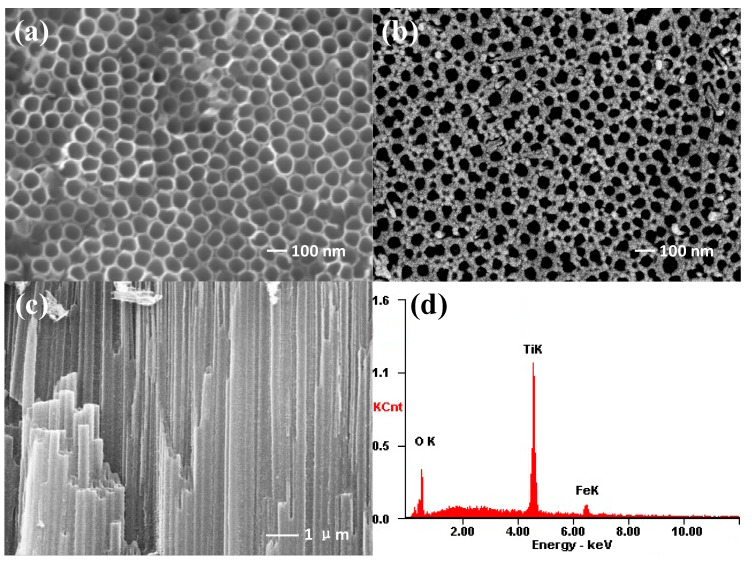
Typical SEM images of (**a**) pure TiO_2_ nanotube arrays without doping, (**b**) Fe^3+^ doped TiO_2_ NTAs, (**c**) cross section of Fe^3+^ doped TiO_2_ NTAs, and (**d**) EDS spectrum of 0.2 mol/L Fe^3+^ doped TiO_2_ NTAs.

**Figure 3 nanomaterials-10-02107-f003:**
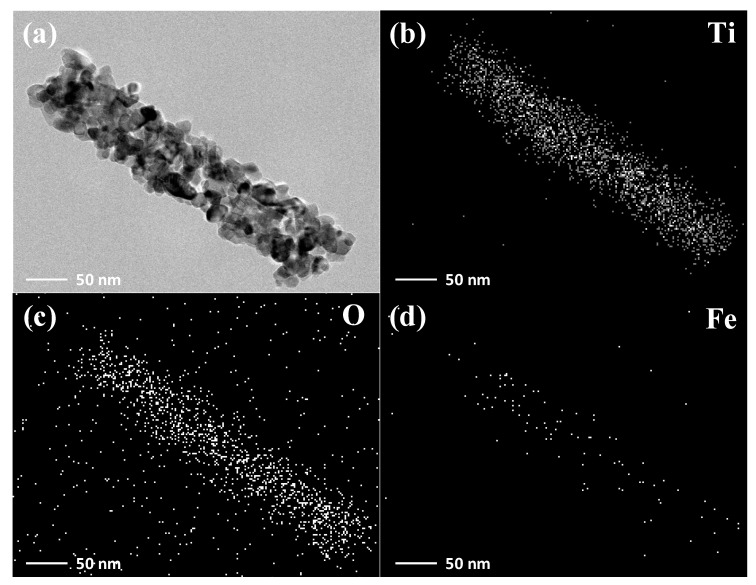
TEM image of 0.2 mol/L Fe^3+^ doped singleTiO_2_ nanotube (**a**), and EDS mapping of Ti (**b**), O (**c**), and Fe (**d**) in the singleTiO_2_ nanotube.

**Figure 4 nanomaterials-10-02107-f004:**
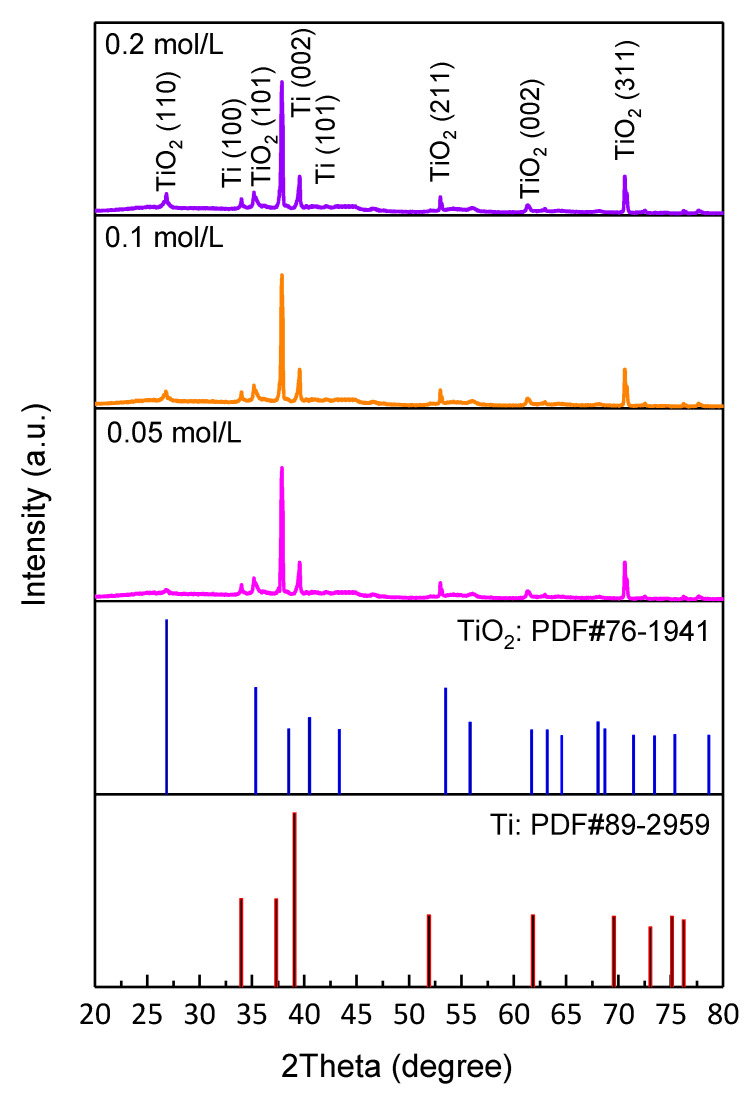
XRD patterns of Fe^3+^ doped TiO_2_ NTAs with various FeCl_3_ solution concentrations and standard PDF cards of rutile phase TiO_2_ and titanium.

**Figure 5 nanomaterials-10-02107-f005:**
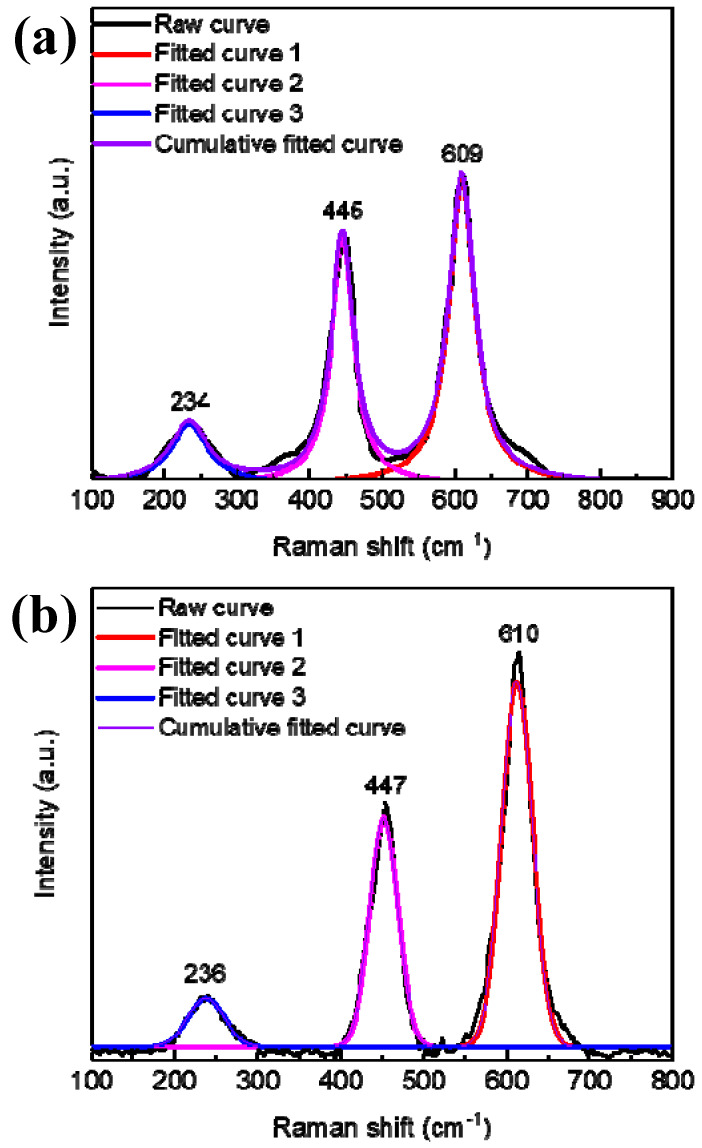
Raman spectra and peak fitting curves of 0.2 mol/L Fe^3+^ doped TiO_2_ NTAs before (**a**) and after (**b**) photocatalytic experiments.

**Figure 6 nanomaterials-10-02107-f006:**
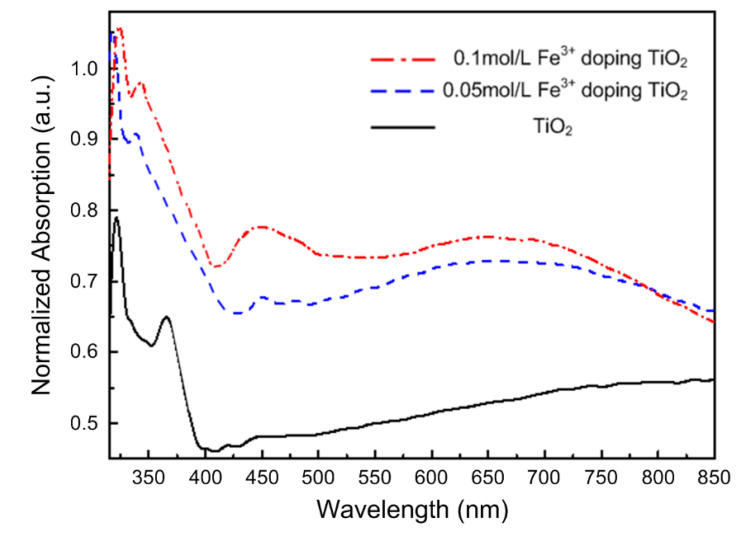
UV-Vis absorption spectra of pure and Fe^3+^ doped TiO_2_ NTAs.

**Figure 7 nanomaterials-10-02107-f007:**
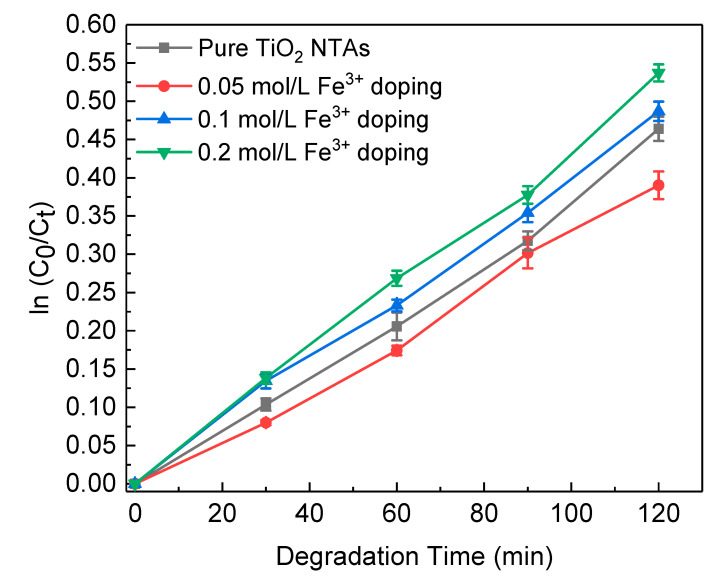
Concentration change of methyl orange (MO) as a function of irradiation time under ultraviolet light for pure and Fe^3+^ doped TiO_2_ NTAs. C_0_, and C_t_ were the concentrations of MO before and after ultraviolet light irradiation, respectively.

**Figure 8 nanomaterials-10-02107-f008:**
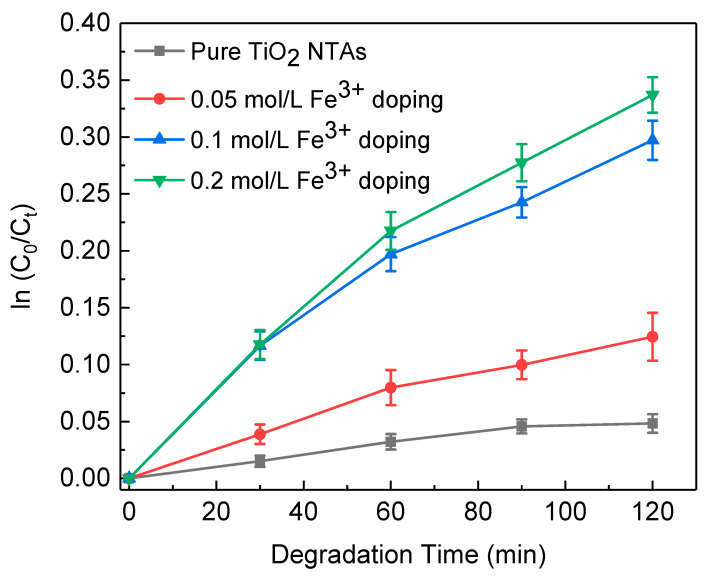
Concentration change of MO as a function of irradiation time under visible light for pure and Fe^3+^ doped TiO_2_ NTAs.

**Table 1 nanomaterials-10-02107-t001:** Weight ratios of Ti, O, and Fe in Fe^3+^ doped TiO_2_ nanotubes arrays (NTAs) with different doping concentrations.

Samples	Ti	O	Fe
0.05 mol/L Fe^3+^	47.64	50.02	2.34
0.1 mol/L Fe^3+^	46.18	49.36	4.46
0.2 mol/L Fe^3+^	40.98	50.49	8.53

**Table 2 nanomaterials-10-02107-t002:** Lattice parameters and interplanar crystal spacing (d values) of pure and Fe^3+^ doped TiO_2_ NTAs samples.

Samples	a = b	c	d Value (Å)				
			(110)	(101)	(211)	(002)	(311)
Pure	4.645	2.992	3.285	2.515	1.706	1.496	1.319
0.05 mol/L Fe^3+^	4.642	2.991	3.282	2.514	1.705	1.496	1.318
0.1 mol/L Fe^3+^	4.640	2.988	3.281	2.512	1.704	1.494	1.317
0.2 mol/L Fe^3+^	4.638	2.987	3.280	2.511	1.704	1.494	1.317

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
