# Peer review of "Preparation of Fe^3+^ Doped High-ordered TiO_2_ Nanotubes Arrays with Visible Photocatalytic Activities"

_nanomaterials, 2020, doi:10.3390/nano10112107_

Round 1

Reviewer 1 Report

It would be nice to have a chemical measure (w/w%) of the amount of Fe in the particles.  However, it clear that Fe help the kinetics.

Author Response

Q: It would be nice to have a chemical measure (w/w%) of the amount of Fe in the particles.  However, it clear that Fe help the kinetics.

A: We are indeed very much grateful to the reviewer for this suggestion, the weight ratio of Ti, O and Fe in these samples were tested by using EDS, and the results were presented in Table 1.

Reviewer 2 Report

The manuscript “Preparation of Fe3+ Doped High-ordered TiO2 Nanotubes Arrays with Visible Photocatalytic Activities“ by J. Zhang et al. reports on the synthesis of iron-ion doped TiO2 nanotubes. XRD and SEM were mainly used to characterise the samples.

The makeup and total quality of the manuscript should be improved before it could be accepted for publication in nanomaterials.

1) The main results claimed by this study seems not to be really new and very closed to previous cited reference 25, and there are more papers related to Fe3+/TiO2 nanotubes (see below).

2) Moreover, the conclusions of this work are already known for a long time. “We find that the Fe3+ doping hardly hinder neither the crystallization of anatase TiO2 .. Photocatalytic testing results indicate the doping of Fe3+ into TiO2 NTAs extends the photo-response range of TiO2 from ultra violet to visible light”.  See:

-Journal of Sol-Gel Science and Technology 21, 109–113, 2001

-Ind. Eng. Chem. Res. 2016, 55, 6619−6633

-Applied Catalysis B: Environmental 129 (2013) 473– 481

Please, cite these references (if necessary).

The authors have to clearly explain the advantages and/or the novelty of this work.

3) The XRD characterisation is not well described. The hkl indices and d values should be added. There are many peaks without any label.

4) The authors have to explain in more details:

-why the intensity of the 101 peak of anatase is not the main peak for pure TiO2 NTs?

-why this peak is broader than others?

-why this peak is totally disappeared after the addition of Fe3+? 

5) The authors have to explain the formula Fe4(TiO3)4. Is it an existing compound? What about Fe2TiO5?

See for instance J. Mater. Chem. A,2014,2,6567–6577 or CrystEngComm,2019,21,34–40.

6) The authors have to perform Raman spectroscopy to improve the characterisation part, especially about the possible formation of Fe2O3.

7) The important point of the release of iron out during the photoirradiation is not well discussed.

Author Response

Q1: The main results claimed by this study seems not to be really new and very closed to previous cited reference 25, and there are more papers related to Fe3+/TiO2 nanotubes (see below).

Q2: Moreover, the conclusions of this work are already known for a long time. “We find that the Fe3+ doping hardly hinder neither the crystallization of anatase TiO2. Photocatalytic testing results indicate the doping of Fe3+ into TiO2 NTAs extends the photo-response range of TiO2 from ultra violet to visible light”. See:

-Journal of Sol-Gel Science and Technology 21, 109–113, 2001

-Ind. Eng. Chem. Res. 2016, 55, 6619−6633

-Applied Catalysis B: Environmental 129 (2013) 473– 481

Please, cite these references (if necessary).

The authors have to clearly explain the advantages and/or the novelty of this work.

Reply to Q1 and Q2:

We are indeed very much grateful to the reviewer for pointing out this issue. According to the reviewer’s request, the advantages of this work has been presented in “Introduction”, and necessary references have also been added.

“Among various transition metal ions, Fe3+ has demonstrated excellent properties when doped with TiO2 NTAs [26, 38, 39]. However, preparation and usage of Fe3+ doped TiO2 NTAs reported in these literatures does not achieve low power consumption, low cost and effective recycling. For example: Sun et al. used high temperature annealing process to prepare TiO2 NTAs [26], Yu et al. adopted high temperature alloy process to fabricate the Fe doped Ti foil [38], Pang et al. prepared discrete Fe3+ doped TiO2 NTAs, which is not conducive to rapid recycling [39].” “In order to facilitate recycling, we proposed a simple and low-cost preparation process of Fe3+ doped TiO2 NTAs on Ti foil under mild environment.”

Q3: The XRD characterisation is not well described. The hkl indices and d values should be added. There are many peaks without any label.

A: We apology for the rough XRD pattern, the XRD pattern has been recomposed, hkl indices of main peaks all so have been added in Figure 3, and the d values have been summarized in Table 2.

Q4: The authors have to explain in more details:

-why the intensity of the 101 peak of anatase is not the main peak for pure TiO2 NTs?

-why this peak is broader than others?

-why this peak is totally disappeared after the addition of Fe3+?

A: We are very sorry that the wrong analysis of XRD data in original manuscript misled you. We reorganized and analyzed the XRD data, and came to new conclusions by comparing the XRD curves with the standard PDF cards. Please see the Figure 3 and line 109 -118 in revised vision.

Q5: The authors have to explain the formula Fe4(TiO3)4. Is it an existing compound? What about Fe2TiO5?

See for instance J. Mater. Chem. A,2014,2,6567–6577 or CrystEngComm,2019,21,34–40.

A: We apology for the error analysis in the original manuscript. Through re-analysis, we found that except for Ti and TiO2, no peaks of other substances were detected in the samples, which also indicated that Fe3+ was doped into the TiO2 nanotubes in the alternative form, and no Fe-containing compounds were formed.

Q6: The authors have to perform Raman spectroscopy to improve the characterisation part, especially about the possible formation of Fe2O3.

A: According to the reviewer’s request, the Raman spectrum of 0.2 mol/L Fe3+ doped sample was presented in Figure 4. After fitting the spectral lines, we found that there were only three Raman peaks of rutile phase TiO2 and no other impurity peaks, which also confirmed the XRD test results that Fe was doped into the lattice of TiO2 and no impurity was formed.

Q7: The important point of the release of iron out during the photoirradiation is not well discussed.

A: Through XRD and Raman test and analysis results, we can see that Fe3+ substituted Ti4+ into the lattice of TiO2 and formed the impurity level, thus expanding the absorption spectrum range of TiO2 nanotubes. Fe3+ did not form other compound impurities, and could not be released in the solution during the photoirradiation.

Reviewer 3 Report

J Zhang et al submitted to nanomaterials an article on TiO2 nanotubes arrays with photocatalytic activities. The paper is interesting but the introduction (see 1- and 2-) and experimental section (see 3- to 5-) needed to be improved. Some comments :

  • 1- The authors claim that diffraction peaks which can be assigned to the anatase phase could be identified in figure 3. I’m not a specialist of XRD, but the presence of rutile could not be discarded (see https://rruff.info/Rutile/R060745). Furthermore, why anatase should be selectively formed under those conditions? The authors should give references and improve their comparison with authentic samples.
  • 2- I could understand that the formation of nanotubes arrays is important for improving the surface, however, the authors should justify the impact of such structure on photocatalytic activity. Each pillar will shadow the pillar nearby ... what did the authors expect.
  • 3- Figure 5 is interesting … but how the absorption is determined. The authors must describe the experimental setup.
  • 4- The degradation of methyl orange is convincing but there no detail on the experiments (Figures 6 and 7). What is the concentration of methyl orange? What is the amount of photocatalyst? What is the intensity of UV or visible light? In both figures, a reaction time of 10 h was claimed in the caption but the degradation time in the abscises was below 120 min.
  • 5- A comparison with a conventional catalyst (P25) under the same conditions is required.

Author Response

Q1: The authors claim that diffraction peaks which can be assigned to the anatase phase could be identified in figure 3. I’m not a specialist of XRD, but the presence of rutile could not be discarded (see https://rruff.info/Rutile/R060745). Furthermore, why anatase should be selectively formed under those conditions? The authors should give references and improve their comparison with authentic samples.

A: We apology for the error XRD analysis in the original manuscript misled you. We reorganized and analyzed the XRD data, and came to new conclusions by comparing the XRD curves with the standard PDF cards. We found that except for Ti and rutile phase TiO2, no peaks of other substances were detected in the samples.

Q2: I could understand that the formation of nanotubes arrays is important for improving the surface, however, the authors should justify the impact of such structure on photocatalytic activity. Each pillar will shadow the pillar nearby ... what did the authors expect.

A: We are grateful for the reviewer’s comments. It has been widely accepted that TiO2 nanotubes are promising for photocatalytic applications due to the following advantages: Firstly, TiO2 nanotubes has excellent electron transport characteristics rather than that it improves the specific surface area. In fact, the specific surface area of TiO2 nanotubes is much smaller than that of TiO2 nanoparticles (Photocatalytic Carbon-Nanotube-TiO2 Composites, Advanced Materials, 2009, 21, 2233-223). Secondly, TiO2 nanotubes array prepared on Ti sheet is easy to be recycled in the process of photocatalysis, while recycling of TiO2 nanoparticle powder must through precipitation and solid-liquid separation, etc. Besides, agglomeration of nano powder is easy to occur during the recycling process, which is not conducive to reuse (Hydrothermal Synthesis of Graphene-TiO2 Nanotube Composites with Enhanced Photocatalytic Activity, ACS Catalysis, 2012, 2.949-956).

Q3: Figure 5 is interesting … but how the absorption is determined. The authors must describe the experimental setup.

A: The absorption spectra were collected by using integrating sphere accessory of the UV−vis spectrometry (Ultrospec 2100 pro), which has been added in experimental section in the revised version.

Q4: The degradation of methyl orange is convincing but there no detail on the experiments (Figures 6 and 7). What is the concentration of methyl orange? What is the amount of photocatalyst? What is the intensity of UV or visible light? In both figures, a reaction time of 10 h was claimed in the caption but the degradation time in the abscises was below 120 min.

A: We apology for the miss matching information between captions and figures. 10 h is low temperature crystallization time of samples and 120 min is for photoactivities test. According to the reviewer’s comments, testing details have been rechecked and added to the revised version as following:

The photocatalytic activities of Fe3+ doped TiO2 NTAs were evaluated on the basis of the degradation of methyl orange (MO) as model organic pollutants in aqueous solutions and measured by the UV vis spectrometer (JASCO V-570 UV/vis/NIR). The surface area (3.0 × 1.6 cm2) of the samples were immersed in the 5×10-5 mol/L MO aqueous solution and irradiated with eight 4W UV bulbs (Toshiba, Black Light Blue, FL4W, Japan), eight 4W visible bulbs (Toshiba, Cool white, FL4W, Japan) with UV filter (>480 nm), for UV photoactivities test and visible photoactivities test, respectively. Before measurement, the samples were soaked for 30 min in dark with magnetic stirring to reach adsorption/desorption equilibrium. The photo-degradation measurement was carried out in an open quartz photoreaction vessel with rapid stirring for 120 min under room temperature. The concentration of the residual MO was measured by the UV-vis spectrometer at about 460 nm.

Q5: A comparison with a conventional catalyst (P25) under the same conditions is required.

A: We are indeed very much grateful to the reviewer for pointing out this issue. If we want to compare the photocatalytic performance of P25 and Fe3+ doped TiO2 NTAs, we must know the exact weight of Fe3+ doped TiO2 nanotubes on each sample, but we could not accurately obtain the accurate weight of Fe3+ doped TiO2 nanotubes on each sample, because TiO2 NTAs were grown on the Ti foil, has a certain binding force, forced stripping will break the morphology of TiO2 nanotubes. In addition, we have compared the photocatalytic performance of undoped and doped samples in this paper, which has been able to explain the photocatalytic performance improvement of Fe3+ doping on the TiO2 nanotube in visible light range.

Reviewer 4 Report

The paper “Preparation of Fe3+ Doped High-ordered TiO2 Nanotubes Arrays with Visible Photocatalytic Activities”

Authors: Jin Zhang, Chen Yang, Shijie Li, Yingxue Xi, Changlong Cai, Weiguo Liu, D. A. Golosov, S. М. Zavadski, S. N. Melniko

In this paper was evaluated possibility to using of a facile method for synthesis iron-ion doped TiO2 nanotubes arrays which uses Fe3+. The Fe3+ ions were simultaneously doped into the TiO2 NTAs during the water-assistant crystallization of amorphous TiO2 NTAs at a temperature as low as 70oC, which indicates the efficient and energy saving merits of this method. The technological conditions and induced photocatalytic performances of the low temperature Fe3+ doping were systematically investigated.

Observation:

The title is clear.

The content is in accord with title.

The manuscript not adheres to the journal's standards

This article contains new aspects; the authors underline the major findings of their work and explain how the use of their proposed materials represents a progress to other similar published papers. Please point more clears the originality.

The Abstract section must be revised; it should refer to the study findings, methodologies, discussion as well as conclusion. Please rewrite the abstract.

The key words permit found article in registers or indexes. Please put in alphabetically.

In the introduction is not clearly described the state of the art of the investigated problem. Please provide references from 2019 and 2020.

The methods are not well described and the equipment and materials were not been adequately described. Please check. In 2.2. Characterization the authors not present adequately devices. Please present details about photocatalysis tests: pH, dosage, time, lamp, initial concentration etc.

The citations not respect guide of authors. Must be in [xx] not superscript or not found!

The paper was written in standard English, more corrections are necessary.

The Conclusion wasn’t been justified, please complete.

Please provide minimum 2 references from this journal (last years), for demonstrated that manuscript is in Nanomaterials topic.

The literature isn’t sufficiently critical, current, and internationally evaluated, this is older, the references from last years are necessary, for demonstrated the actuality.

The size of the article is appropriate to the contents.

It is a cleaner paper about synthesis, characterization and application of new materials, with the text presented and arranged clearly and concisely.

The manuscript isn’t in format, references not respect guide for authors.

Even though the authors synthesized catalyst systems with an intended weight% loading, they should validate this loading

What is the recyclability of these systems? One synthetic method may have an initial enhanced reactivity,

For better understating of the readers, please use clearer figures.

For understanding of the oxidation state and morphology of the synthesized catalysts, authors should perform more analysis of the catalysts, and, discuss in the manuscript.

The doping have a harmful effect on the photocatalytic activity of TiO2. Why in the case of 0.05 mol/L Fe3+ this effect not appear (Fig. 6)?  Has the shadowing effect of the dopant on the TiO2 active sites?

Please respect guide of authors and verify all references. The references [27] is write in 0? Journal there are abbreviated or not… etc.

Author Response

Q1: The title is clear.

A: Thanks for the reviewer's affirmation.

Q2: The content is in accord with title.

A: Thanks for the reviewer's positive evaluation.

Q3: The manuscript not adheres to the journal's standards

A: Based on the journal's standards, the manuscript has been reformatted.

Q4: This article contains new aspects; the authors underline the major findings of their work and explain how the use of their proposed materials represents a progress to other similar published papers. Please point more clears the originality.

A: According to the reviewer’s request, the originality of this work has been presented in “Introduction”:

“Among various transition metal ions, Fe3+ has demonstrated excellent properties when doped with TiO2 NTAs [26, 38, 39]. However, preparation and usage of Fe3+ doped TiO2 NTAs reported in these literatures does not achieve low power consumption, low cost and effective recycling. For example: Sun et al. used high temperature annealing process to prepare TiO2 NTAs [26], Yu et al. adopted high temperature alloy process to fabricate the Fe doped Ti foil [38], Pang et al. prepared discrete Fe3+ doped TiO2 NTAs, which is not conducive to rapid recycling [39].” “In order to facilitate recycling, we proposed a simple and low-cost preparation process of Fe3+-doped TiO2 NTAs on Ti foil under mild environment.”

Q5: The Abstract section must be revised; it should refer to the study findings, methodologies, discussion as well as conclusion. Please rewrite the abstract.

A: According to the reviewer’s request, the abstract has been rewritten.

Q6: The key words permit found article in registers or indexes. Please put in alphabetically.

A: The order of key words has been readjusted in alphabetically.

Q7: In the introduction is not clearly described the state of the art of the investigated problem. Please provide references from 2019 and 2020.

A: According to the reviewer’s request, the newest references (ref. [1], [2], [5], [7], [28] and [35]) have been added in revised vision.

Q8: The methods are not well described and the equipment and materials were not been adequately described. Please check. In 2.2. Characterization the authors not present adequately devices. Please present details about photocatalysis tests: pH, dosage, time, lamp, initial concentration etc.

A: We are very sorry for our previous negligence; the necessary equipment and experimental details have been added in “Materials and Methods” section.

Q9: The citations not respect guide of authors. Must be in [xx] not superscript or not found!

A: According to the guide of authors, the citations have been reformatted.

Q10: The paper was written in standard English, more corrections are necessary.

A: The English languish has now been carefully polished in the revised version as possible as we can. If there is any more suggestion for its betterment further, we will certainly welcome all those for implementation with all appreciation from us.

Q11: The Conclusion wasn’t been justified, please complete.

A: According to the reviewer’s request, the “Conclusions” has been rewritten in revised vision.

Q12: Please provide minimum 2 references from this journal (last years), for demonstrated that manuscript is in Nanomaterials topic.

A: Following to the reviewers' advice, 4 references ([1], [2], [5] and [7]) related to TiO2 nanomaterials from this journal (newest) have been added in the revised version.

Q13: The literature isn’t sufficiently critical, current, and internationally evaluated, this is older, the references from last years are necessary, for demonstrated the actuality.

A: According to the latest literature report, we have added relevant content in "Introduction", and proposed the advantages of this paper that are different from other literatures. The highlights of this paper lie in the preparation method of Fe3+ doped TiO2 NTAs with low power consumption and low cost, as well as the easy recovery characteristics different from other TiO2 nano powder catalysts.

Q14: The size of the article is appropriate to the contents.

A: Thanks very much for your comments, we have polished our manuscript again with the best effort of us.

Q15: It is a cleaner paper about synthesis, characterization and application of new materials, with the text presented and arranged clearly and concisely.

A: We are indeed very much grateful to the reviewer for positive evaluation of this manuscript.

Q16: The manuscript isn’t in format, references not respect guide for authors.

A: According to the reviewers' requirement, the whole manuscript has been reformatted and the references have been rearranged as required.

Q17: Even though the authors synthesized catalyst systems with an intended weight% loading, they should validate this loading

A: Based on reviewer’s suggestion, the mass ratio of Ti, O and Fe in these samples were tested by using EDS, and the results were presented in Table 1.

Q18: What is the recyclability of these systems? One synthetic method may have an initial enhanced reactivity,

A: After ten times reusing, recycling tests of the photocatalytic activity of the best sample doped by 0.2 mol/L Fe3+ were also carried out under the same condition, and negligible deteriorate were obtained. It indicates that our systems are very stable and the recyclability is good. The recycling results were provided as error bars in Fig. 6 and Fig. 7.

Q19: For better understating of the readers, please use clearer figures.

A: Thanks very much for the reviewer’s suggestion, more clearer figures have been updated in the revised version.

Q20: For understanding of the oxidation state and morphology of the synthesized catalysts, authors should perform more analysis of the catalysts, and, discuss in the manuscript.

A: According to the reviewer’s comments, we have performed EDS, Raman, SEM section view, and photocatalytic recycling test with the synthesized samples, and the results and corresponding discussions have now added to the revised version, including Figure 2, Figure 4, Figure 6, Figure 7 and Table 1.

Q21: The doping have a harmful effect on the photocatalytic activity of TiO2. Why in the case of 0.05 mol/L Fe3+ this effect not appear (Fig. 6)? Has the shadowing effect of the dopant on the TiO2 active sites?

A: We agree with the reviewer that in certain circumstances the doping would have a harmful effect on the photocatalytic activity of TiO2. In this study, the case of 0.05 mol/L Fe3+ would harm its photocatalytic activity under UV irradiation. This is might due to that the UV light has provided enough energy to generated electron-hole pairs, and a small amount doping of Fe3+ tend to act a recombination center, which will decrease the photoactivities. With increase doping content or under visible light irradiation, the major effect of the Fe3+ doping is to introduce impurity levels in the band gap of TiO2, and become a shallow trap of photogenerated electrons or holes, for reducing the recombination of electron-hole pairs. (Ghicov, A; Schmidt, B; Kunze, J; Schmuki, P, Photoresponse in the visible range from Cr doped TiO2 nanotubes. Chemical Physics Letters 2007, 433 (4-6), 323-326. Shen, X. Z; Guo, J; Liu, Z. C; Xie, S. M, Visible-light-driven titania photocatalyst co-doped with nitrogen and ferrum. Applied Surface Science 2008, 254 (15), 4726-4731.)

Q22: Please respect guide of authors and verify all references. The references [27] is write in 0? Journal there are abbreviated or not… etc.

A: We are very sorry for this mistake; the references have been corrected in revised version.

Round 2

Reviewer 2 Report

Thanks to the authors for some improvements.

Nevertheless, there is still some points to be clarified:

1) The authors have to explain how Ti4+ could be replaced by Fe3+ only at 70°C. EDX mapping images of the NTs should be performed in order to see if some amorphous Fe hydroxide or oxide layer is formed. At least, a gradient of Fe3+ should be found.

 2) This conclusion of this sentence is wrong and it should be removed or corrected, "In addition, there is no other impurity peaks can be observed, which indicates that Fe3+ is doped into the TiO2 nanotubes 124 in the alternative form, and no Fe-containing compounds are formed." XRD is not at all a good technic for that, amorphous phases or thin layer cannot be detected.

2) Still, the important point of the release of iron out during the photoirradiation is not well discussed. Raman and EDX analysis should be compared before and after the photocatalytic experiments.

Author Response

Q1: The authors have to explain how Ti4+ could be replaced by Fe3+ only at 70°C. EDX mapping images of the NTs should be performed in order to see if some amorphous Fe hydroxide or oxide layer is formed. At least, a gradient of Fe3+ should be found.

A: It is believed that the crystallization of amorphous TiO2 will proceed via a dissolution-precipitation process even under room temperature(25oC), during which randomly distributed TiO62-octahedra are dissolved and rearrange themselves driven by water, and then precipitate starts nucleating. (D. Wang, L. Liu, F. Zhang, K. Tao, E. Pippel, K. Domen, Spontaneous phase and morphology transformations of anodized titania nanotubes induced by water at room temperature. Nano Lett. 11(9), 3649–3655 (2011). https://doi.org/10.1021/nl2015262) For this study, accompanying the dissolution-precipitation, the Fe3+ ions from aqueous solution would be involved into the TiO2 crystals, which might explain that the doping of Fe3+ into rutile phase TiO2 nanotubes could happened as low as 70oC.

According to the reviewer’s request, TEM and EDS mapping were employed to explain the distribution of Fe in TiO2 nanotubes. Please see Figure 3 and line 110-115.

Q2: This conclusion of this sentence is wrong and it should be removed or corrected, "In addition, there is no other impurity peaks can be observed, which indicates that Fe3+ is doped into the TiO2 nanotubes in the alternative form, and no Fe-containing compounds are formed." XRD is not at all a good technic for that, amorphous phases or thin layer cannot be detected.

A: We are indeed very much grateful to the reviewer for pointing out this issue. This sentence has been modified to “In addition, no other impurity peaks are found, and whether Fe3+ is doped into TiO2 nanotubes in alternative forms needs further discussion.”.

Q3: Still, the important point of the release of iron out during the photoirradiation is not well discussed. Raman and EDX analysis should be compared before and after the photocatalytic experiments.

A: According to reviewer’ s request, the Raman and EDX analysis have been compared before and after the photocatalytic experiments in revised manuscript. Results show that Fe3+ did not form other compound impurities and could not be released in the solution during the photoirradiation. Please see Figure 5(b) and line 160-167.

Reviewer 3 Report

The authors have corrected their paper according to the reviewer's comments;

The paper can be accepted.

Author Response

Thanks for the reviewer's affirmation.

Reviewer 4 Report

The authors modified manuscript in accordance with recommendations.

Author Response

Thanks for the reviewer's positive evaluation.